# Biologic Treatment Options for Pediatric Psoriasis and Atopic Dermatitis

**DOI:** 10.3390/children6090103

**Published:** 2019-09-11

**Authors:** Abigail Cline, Gregory J. Bartos, Lindsay C. Strowd, Steven R. Feldman

**Affiliations:** 1Center for Dermatology Research, Department of Dermatology, Wake Forest School of Medicine, 475 Vine St, Winston-Salem, NC 27101, USA; 2Department of Dermatology, Wake Forest School of Medicine, 475 Vine St, Winston-Salem, NC 27101, USA; 3Department of Pathology, Wake Forest School of Medicine, 475 Vine St, Winston-Salem, NC 27101, USA; 4Department of Social Sciences & Health Policy, Wake Forest School of Medicine, Winston-Salem, NC 27101, USA; 5Department of Dermatology, University of Southern Denmark, Campusvej 55, 5230 Odense, Denmark

**Keywords:** children, adolescents, biologics, atopic dermatitis, psoriasis

## Abstract

*Background and Objectives:* Severe, recalcitrant cases of pediatric psoriasis or atopic dermatitis may necessitate treatment with biological agents; however, this may be difficult due to lack of treatment options and standardized treatment guidelines. This review evaluates the biological treatment options available, including off-label uses, and provides a basic therapeutic guideline for pediatric psoriasis and atopic dermatitis. *Materials and Methods:* A PubMed review of biological treatments for pediatric psoriasis and atopic dermatitis with information regarding age, efficacy, dosing, contra-indications, adverse events, and off-label treatments. *Results:* Currently there are three European Medicines Agency (EMA)-approved biological treatment options for pediatric psoriasis: etanercept, ustekinumab, and adalimumab. While dupilumab was recently Food and Drug Administration (FDA)- and EMA-approved for adult atopic dermatitis, it is still not yet approved for pediatric atopic dermatitis. *Conclusions:* Given the high morbidity associated with pediatric atopic dermatitis and psoriasis, there is a need for more treatment options. Further research and post-marketing registries are needed to extend the use of biologics into pediatric patients.

## 1. Introduction

Dermatologic conditions in pediatric populations, such as atopic dermatitis (AD) and psoriasis (Ps), put children at an increased risk for low self-esteem, depression, anxiety, social isolation, and suicidal ideation [1,2,3]. These diseases impact a patient’s quality of life, and thus warrant early recognition and treatment to decrease their risk of physical and psychologic morbidity. Although caregivers may prefer the risk profile of topical therapies compared to systemic agents, topical products are often inadequate to effectively treat moderate-to-severe AD or Ps. Frequent application of topical medication to large areas of skin is difficult, expensive, and increases the risk of adverse effects (AEs). Moderate-to-severe AD or Ps may warrant systemic treatment with traditional systemic medications, small molecule inhibitors, or biological agents.

Biologics are an attractive option because of their convenient dosing regimens, low frequency of laboratory monitoring, and high efficacy in adults. While the number of biologic agents to treat dermatologic conditions in adult patients has expanded in the past decade, biologic options available for pediatric AD and Ps remain limited. Physicians may consider prescribing these treatments off-label; however, off-label prescription may result in issues with insurance coverage. Therapeutic guidelines are lacking and further research is needed to evaluate dosing, safety, tolerability, and treatment efficacy in pediatric populations. This literature review will summarize and assess the use of biologic agents used in dermatology to treatment AD and Ps. We hope this review will inform providers on the indications, safety, and treatment efficacy of biologic therapies in pediatric and adolescent patients with AD or Ps.

## 2. Methods 

A PubMed search included key words “psoriasis”, “psoriatic”, “atopic dermatitis”, “eczema”, “pediatric”, “child”, or “children” together with “treatment”, “off-label”, “etanercept”, “infliximab”, “adalimumab”, “ustekinumab”, “secukinumab”, “certolizumab pegol”, “guselkumab”, “tildrakizumab”, “brodalumab”, “ixekizumab”, “dupilumab”, “risankizumab”, “Food and Drug Administration” (FDA), or “European Medicines Agency” (EMA). Data regarding age, efficacy, dosing regimens, frequency, contraindications, adverse events, and off-label treatments were recorded.

## 3. Approved Biologics for Pediatric Populations

### 3.1. Dupilumab

Dupilumab is a fully human monoclonal antibody targeting the α subunit shared by interleukin (IL)-4 and IL-13 receptors, thereby inhibiting their downstream signaling [4]. Dupilumab has improved AD in adults with moderate-to-severe AD compared to placebo, and it is generally well-tolerated with less concern for toxicity compared to other systemic medications used for AD [5]. It is EMA-approved for treating adult patients with AD, and FDA-approved for adult and adolescent patients (age 12 years and older) with AD (Table 1). 

A randomized, double-blind, placebo-controlled trial evaluated dupilumab in 251 adolescents (age 12 to 17 years) with moderate-to-severe AD for 16 weeks, followed by a 12-week open-label extension. Patients received either dupilumab 200 mg (body weight < 60 kg) or 300 mg (body weight > 60 kg) every 2 weeks, dupilumab 300 mg every 4 weeks, or placebo. All patients randomized to dupilumab received either 400 mg (body weight < 60 kg) or 600 mg (body weight > 60 kg) as a loading dose. At week 16, more dupilumab patients (either every 2 weeks or every 4 weeks) than placebo patients achieved a 50% improvement in their Eczema Area and Severity Index (EASI 50) results (61% and 54.8% vs 12.9, *P* < 0.001) and showed better EASI 75 results (41.5% and 38.1% vs 8.2%, *P* < 0.001), along with better results in an Investigator Graded Assessment (IGA) of clear or almost clear (24.2% and 17.9% vs 2.4%, *P* < 0.001). Adolescent patients receiving dupilumab reported improved signs and symptoms of AD (including pruritus) and quality of life. For most endpoints, the two week regimen was superior to the four week regimen [6,7]. AEs were similar across all treatment arms, and the one treatment-emergent AE that lead to discontinuation was in the placebo group. The dupilumab groups reported higher rates of conjunctivitis and injection site reactions, whereas the placebo group reported higher rates of AD exacerbation and non-herpetic skin infections [7]. These adverse events were similar to those described in the adult dupilumab clinical trials. 

A retrospective chart review identified six pediatric patients (ages 7 to 15) treated with dupilumab every two weeks. Patients > 40 kg received 300 mg and patients < 40kg received 150 mg. All patients receiving dupilumab had at least a 2-point decrease in IGA, with an average treatment duration of 8.5 months (range: 6–11). After treatment, three patients (50%) had an IGA of clear or almost clear. One patient discontinued therapy after 6 months, when her skin was reportedly almost clear. Within 2 months of discontinuing treatment, her skin worsened to an IGA of 3 and she subsequently restarted dupilumab. No side effects were reported [8].

Clinical trials are currently underway to evaluate dupilumab in pediatric patients age 6 months to 6 years, ages 6 to 18, and ages 6–12 years with co-administration of topical corticosteroids [9,10,11].

### 3.2. Etanercept

Etanercept is FDA approved for pediatric Ps in patients 4 years or older and EMA approved for pediatric Ps in patients 6 years or older (Table 1) [12]. Consisting of the fragment crystallizable (Fc) portion of IgG1 fused with a recombinant human tumor necrosis factor (TNF) receptor protein, etanercept binds to both soluble and membrane-bound forms of TNFα. Etanercept is an FDA- and EMA-approved treatment for plaque Ps, psoriatic arthritis, rheumatoid arthritis, juvenile rheumatoid arthritis, and ankylosing spondylitis. 

A 48-week, double-blind trial randomized 211 pediatric patients with Ps (4–17 years of age) to either once-weekly subcutaneous injections of placebo or etanercept (0.8 mg/kg with a maximum of 50 mg) for 12 weeks, followed by 24 weeks of once-weekly open-label etanercept. From week 36 to week 48, 138 patients were randomized a second time to either placebo or etanercept. A 75% or greater improvement from baseline in the Psoriasis Area and Severity Index (PASI) at week 12 was the primary endpoint. At week 12, more patients receiving etanercept than patients receiving placebo achieved acceptable results for PASI 50 (75% vs. 23%, *P* < 0.001), PASI 75 (57% vs 11%, *P* < 0.001), PASI 90 (27% vs. 7%, *P* < 0.001), and a Physician’s Global Assessment (PGA) of clear or almost clear (53% vs. 13%, *P* < 0.001). At week 36, after 24 weeks of treatment with open-label etanercept, 68% of patients initially receiving etanercept and 65% of patients initially receiving placebo achieved PASI 75. During the withdrawal period, 42% of the patients assigned to placebo at the second randomization lost treatment response. During the open-label etanercept treatment, four serious AEs (including three infections) occurred in three patients, all of which resolved without complication [13].

A 5-year open-label extension study evaluated the use of etanercept (0.8 mg/kg) for treating Ps in 181 pediatric patients (4–17 years of age). End points included occurrence of AEs and serious AEs, PASI 75, PASI 90, and PGA. By week 264, 161 (89.0%) patients had reported an AE, with the most common ones being upper respiratory tract infection (37.6%), nasopharyngitis (26.0%), and headache (21.5%). Although seven patients reported 8 serious AEs, researchers considered only 1 (cellulitis) to be treatment-related. There were no opportunistic infections or malignancy reported. The percentage of patients achieving PASI 75 (60–70%), PASI 90 (30–40%), and PGA status of clear or almost clear (40–50%) were maintained through week 264. During the study, 10.7% of patients tested positive for anti-etanercept antibodies; however, none of the anti-etanercept antibodies were neutralizing [14].

Given the quality and quantity of data, providers should use etanercept as a first-line agent for systemic treatment of moderate-to-severe pediatric Ps in patients 4 years and older. The treatment dose for pediatric patients is 0.8 mg/kg weekly, with a maximum dose of 50 mg. Etanercept is associated with fewer AEs than the traditional systemic agents methotrexate or acitretin [15]. Common AEs include upper respiratory tract infection, pharyngitis, injection site reactions, and headaches. Serious AEs include cellulitis. There have been no cases of opportunistic infections or malignancy reported in children or adolescents [14]. Although TNF inhibitors carry black box warnings for increased risk of lymphoma and other malignancies in pediatric populations, a clear relationship between the drug and malignancy has not been established [16].

### 3.3. Ustekinumab

Ustekinumab is FDA-approved and EMA-approved for treatment of moderate-to-severe Ps in adolescents (12 years and older) (Table 1) [17]. Consisting of a fully human monoclonal antibody targeting the p40 protein subunit shared by IL-12 and IL-23, ustekinumab inhibits both the Th1 and Th17 pathways [18]. Dosing for ustekinumab is weight-based. Children <60 kg should receive 0.75 mg/kg, children between 60 kg and 100 kg should receive 45 mg, and children >100 kg should receive 90 mg. Loading doses of ustekinumab are administered at weeks 0 and 4, and then maintenance dosing is administered every 12 weeks. This convenient dosing regimen makes ustekinumab appealing for pediatric populations. 

A randomized, controlled trial assigned 110 Ps patients age 12–17 years to ustekinumab standard dosing (SD; 0.75 mg/kg (≤60 kg), 45 mg (>60–≤100 kg), and 90 mg (>100 kg)) or half-standard dosing (HSD; 0.375 mg/kg (≤60 kg), 22.5 mg (>60–≤100 kg), and 45 mg (>100 kg)) at weeks 0, 4, and then every 12 weeks; or placebo at weeks 0 and 4 with crossover to ustekinumab SD or HSD at week 12. Researchers monitored for AEs through week 60. By week 12, more ustekinumab patients receiving SD or HSD than placebo achieved PASI 75 (80.6% and 78.4% vs. 10.8%, *P* < 0.001), PASI 90 (61.1% and 54.1% vs. 5.4%, *P* < 0.001), and PGA of 0 (clear) or 1 (almost clear) (69.4% and 67.6% vs. 5.4%, *P* < 0.001). PASI responses observed at week 12 were sustained at week 52 among patients continuing ustekinumab. Participants randomized to placebo and later crossed over to the SD achieved better PASI responses than those randomized to placebo and then crossed over to the HSD, with the following results: PASI 75 (100% vs 70.6%), PASI 90 (94.1% vs. 52.9%), PGA of clear or almost clear (94% vs. 68%) [17].

The standard ustekinumab dose provided treatment responses comparable to those reported in adults. At week 12, 56.8% of placebo patients, 51.4% of HSD patients, and 44.4% of SD patients reported at least one AE. Through week 60, 81.8% of participants reported AEs. AEs in adolescent patients were consistent with those in adult patients receiving ustekinumab. The most common AEs were nasopharyngitis (34.5%), upper respiratory tract infection (12.7%), and pharyngitis (8.2%). No cases of opportunistic infections or malignancy were reported [17]. There were no differences between patients receiving ustekinumab and patients receiving placebo in terms of short-term and longer-term AEs and few participants withdrew because of adverse effects. Given the efficacy, safety data, and convenient dosing schedule, providers should consider ustekinumab as a first-line treatment for adolescent patients with moderate-to-severe Ps. 

### 3.4. Adalimumab

Adalimumab is a fully human anti-TNFα monoclonal IgG1 antibody that binds to soluble and membrane-bound TNFα. The FDA has approved adalimumab for the treatment of plaque Ps, ankylosing spondylitis, psoriatic arthritis, hidradenitis suppurativa, juvenile arthritis, ulcerative colitis, and Crohn’s disease. While adalimumab is FDA-approved for children with Crohn’s disease (6 years or older) and Juvenile Idiopathic Arthritis (JIA) (2 years or older), it is not FDA-approved for children with Ps (Table 1). The EMA has approved adalimumab in children 4 years or older with Ps unresponsive to topical treatments and phototherapy. 

A multicenter, randomized, controlled trial compared adalimumab to methotrexate in 114 pediatric patients with Ps (4–17 years of age). The study had four periods: (1) a double-blind, randomized, controlled trial for 16 weeks; (2) an observational study of treatment withdrawal for up to 36 weeks; (3) a double-blind retreatment study for 16 weeks; and (4) a long-term follow-up period for up to 52 weeks. Following randomization, participants were randomized to either SD adalimumab (0.8 mg/kg up to a maximum of 40 mg, and then 0.8 mg/kg every other week), HD adalimumab (0.4 mg/kg up to a maximum of 20 mg, and then 0.4 mg/kg every other week), or methotrexate (0.1 mg/kg up to a maximum of 7.5 mg, and then a weekly dose of up to 0.4 mg/kg, up to a maximum dose of 25 mg/week). At week 16, more patients receiving SD adalimumab achieved PASI 75 compared to HD adalimumab and methotrexate (58% vs. 44% vs. 32%, respectively) and a PGA score of 0 (clear) or 1 (almost clear) (61% vs. 41% vs. 41%, respectively). Following the first period, patients entered the withdrawal period of up to 36 weeks. While treatment withdrawal in responders resulted in loss of disease control in many patients, adalimumab re-treatment successfully recovered therapeutic response in most patients [19]. 

Adalimumab is more effective than methotrexate while having a similar safety profile. Overall, 74% of the trial participants reported AEs during the first treatment period: 68% receiving SD adalimumab, 77% receiving HD adalimumab, and 76% receiving methotrexate. Infections were the most frequently reported AEs (45% reported in the SD adalimumab, 56% in the HD adalimumab, and 57% in the methotrexate group). No malignancies and no deaths related to study drugs were reported. Investigators considered all serious AEs to be unrelated or probably unrelated to the study drugs, except for one case of eye nevus in a participant receiving 0.8 mg/kg of adalimumab. 

Adalimumab is efficacious, well-tolerated, and safe in pediatric Ps. The treatment dose for pediatric patients is 0.8mg/kg, with a maximum dose of 40mg given at weeks 0 and 1, and then every 2 weeks.

## 4. Off-Label

Infliximab is FDA-approved for children 6 years or older with Crohn’s disease, but not for Ps [20]. While infliximab effectively treats Crohn’s disease in pediatric populations and Ps in adult populations, it can paradoxically induce Ps [21]. Infliximab use in pediatric populations is limited to case reports and anecdotal experience, most often in patients with severe pustular Ps [22,23,24]. The dose for pediatric patients with Ps is 3.3–5 mg/kg administered at weeks 0, 2, and 6, and then every 7–8 weeks thereafter. Infliximab is a useful rescue treatment owing to its efficacy and quick onset of action [23]. Patients should avoid sporadic use as this can induce neutralizing antibodies, which decrease efficacy and increase risk of transfusion reactions [23]. Common AEs include uncomplicated infections and headache. Serious AEs include infusion reactions, hepatotoxicity, cytopenias, and malignancies. In pediatric patients treated with infliximab, higher rates of malignancies have been reported. However, firm conclusions cannot be made due to confounding factors such as concomitant immunosuppressive medications and underlying disease cancer risk [25]. 

Certolizumab pegol is FDA-approved for Ps and Psoriatic Arthritis (PsA) in adults, but it is not FDA-approved or EMA-approved for pediatric use. Certolizumab pegol effectively treats pediatric patients with JIA with a safety profile similar to other TNF inhibitors. The plasma concentrations of certolizumab pegol in pediatric patients fall largely within the range seen in adults [26]. Clinical trials are currently investigating cerolizumab pegol in pediatric arthritis and Crohn’s disease, but not in Ps [27,28]. 

Secukinumab is FDA-approved for Ps and PsA, but it is not FDA-approved or EMA-approved for pediatric use for any medical condition. Two pediatric patients with Ps due to interleukin-36 receptor antagonist (DITRA) deficiency showed improvement with secukinumab treatment [29,30]. In one case, a 4-year-old boy with generalized pustular Ps achieved complete clearance with secukinumab 75 mg/week [29]. In the second case, an adolescent boy cleared generalized erythrodermic Ps with secukinumab 150 mg/week. Doses were 150mg at week 1, 2, 3, and 4, and then every 4 weeks thereafter. At 1-year follow-up, the patient remained recurrence-free [30]. Clinical trials are currently evaluating the safety and efficacy of secukinumab in pediatric patients with Ps [31,32].

Other biologics currently FDA- and EMA-approved for adults with Ps that are undergoing clinical trials for pediatric Ps include ixekizumab [33], guselkumab [34], and brodalumab [35]. Risankizumab is currently undergoing clinical trials in adolescents with AD [36].

## 5. Discussions

The treatment of pediatric AD and Ps is complicated as there are limited data regarding efficacy and safety of treatment options. Several factors must be considered while choosing the appropriate biologic therapy for Ps, and key considerations for the pediatric population include dosing schedule, approval for pediatric population, and safety profile [37]. General treatment guidelines stem from sensible interpretation of our knowledge on available therapies and recommendations from experienced practitioners. Traditional agents are not appropriate for long-term use in the pediatric population due to the risk of side effects. Biologics provide important therapeutic options for many patients with moderate-to-severe AD or Ps who have not had an adequate response to other treatments. Dupilumab is safe and effective in adolescent populations; however, there is less safety data in children. FDA-approved etanercept and ustekinumab and EMA-approved adalimumab are first-line systemic therapies for moderate-to-severe pediatric Ps. Currently, the three biologics have no head-to-head trials evaluating their efficacy and safety in pediatric patients. The safety and efficacy profile observed over decades of use and the data to be elucidated in the future should comfort parents worried about systemic effects [38]. Our analysis is in agreement with recently published recommendations for Ps, which concluded that etanercept and ustekinumab are preferred agents for the treatment of pediatric Ps [39]. When considering dupilumab, etanercept, ustekinumab, or adalimumab, pediatric patients should have received all required vaccinations. 

Multiple clinical trials are underway for the treatment of pediatric AD and Ps that will hopefully provide more effective options to treat these chronic conditions [40,41]. As future research elucidates the pathophysiologies of AD and Ps, new targets will be identified and new treatments will be developed. Biologics may be the most effective treatment option for moderate-to-severe AD and Ps, but they do have complications. Biologics are costly, require repeated injections (which can be difficult in pediatric patients), and are not curative. If treatment is discontinued or patients have poor adherence, disease relapse or rebound can occur [8,42].

As more biologic treatment options become available for both AD and Ps, there is an increased need for patient monitoring. Post-market registries are necessary to evaluate long-term AEs of pediatric patients who may be on biologics for decades. Furthermore, the occurrence of most comorbidities increases with age. Patients with Ps often have multiple comorbidities, including cardiovascular disease, diabetes type II, depression, and autoimmune disease [43]. As these pediatric patients age, they may develop comorbidities associated and not associated with AD and Ps. The effects of prolonged use of biologics on these comorbid conditions will become more clinically relevant.

## Figures and Tables

**Table 1 children-06-00103-t001:** Food and Drug Administration (FDA)- and European Medicines Association (EMA)-approved biologic treatments in pediatric patients with atopic dermatitis or psoriasis.

Medication	Mechanism	Dosing Regimen	Contraindications	Adverse Effects	Baseline Lab Monitoring	FDA Approval	EMA Approval
**Dupilumab**	α subunit of IL-4 and IL-13 receptors	For patients < 60 kg: SC injection of 400mg at week 0 and then 200mg every other week	Hypersensitivity	Common: injection site reactions, conjunctivitis, herpes infections, upper respiratory injection	-	Treatment of atopic dermatitis in patients >12 years	Treatment of asthma in patients >12 years
		For patients >60 kg: SC injection of 600 mg at week 0 and then 300 mg every other week		Severe: inflammation of blood vessels		Treatment of asthma in patients >12 years	
**Etanercept**	TNF-α inhibitor	SC injection of 0.8 mg/kg per week with a maximum of 50 mg per week	Active infections, history of heart failure, multiple sclerosis, demyelinating disease	Common: upper respiratory infections, pharyngitis, injection site reactions, headaches	PPD or QuantiFERON-gold, CBC, CMP, hepatitis Panel	Treatment of psoriasis in patients >4 years of age	Treatment of psoriasis in patients >6 years of age
						Treatment of juvenile idiopathic arthritis in patients >2 years of age	Treatment of psoriatic arthritis >12 years of age
				Severe: cellulitis		Treatment of Crohn’s disease in patients >6 years of age	Treatment of juvenile idiopathic arthritis in patients >2 years of age
**Adalimumab**	TNF-α inhibitor	For patients 15 kg to <30 kg: SC injection of 20 mg at week 0 and then 20 mg every other week	Active infections, history of heart failure, multiple sclerosis, demyelinating disease	Common: upper respiratory infections, uncomplicated infections, injection site reactions	PPD or QuantiFERON-gold, CBC, CMP, Hepatitis Panel	Treatment of juvenile idiopathic arthritis in patients >2 years of age	Treatment of psoriasis in patients >4 years of age
						Treatment of Crohn’s disease in patients >6 years of age	Treatment of juvenile idiopathic arthritis in patients >2 years of age
		For patients >30 kg: SC injection of 40 mg at week 0 and then 40 mg every other week		Severe: cellulitis, hyperlipidemia		Treatment of uveitis in patients >2 years of age	Treatment of Crohn’s disease in patients >6 years of age psoriasis
						Treatment of hidradenitis suppurativa in patients >12 years of age	Treatment of uveitis in patients >2 years of age
**Ustekinumab**	p40 subunit of IL-12 and IL-23	For patients <60kg: SC injection of 0.75 mg/kg at week 0, week 4, and then every 12 weeks	Active infections, history of recurrent infections	Common: upper respiratory infection, headache, injection site reaction	PPD or QuantiFERON-gold	Treatment of psoriasis in patients >12 years	Treatment of psoriasis in patients >12 years
		For patients 60 kg to 100 kg: SC injection of 45 mg at week 0, week 4, and then every 12 weeks					
		For >100kg: SC injection of 90 mg at week 0, week 4, and then every 12 weeks		Severe: skin carcinoma			

Abbreviations: IL = interleukin; SC = subcutaneous; TNF = tumor necrosis factor; PPD = purified protein derivative; CBC = complete blood count; CMP = complete metabolic panel; BSA = body surface area.

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
