# Peer review of "Biologic Treatment Options for Pediatric Psoriasis and Atopic Dermatitis"

_children, 2019, doi:10.3390/children6090103_

Round 1
Reviewer 1 Report
The article summarizes the use of biological therapies in atopic dermatitis and psoriasis in the pediatric population. It strengthens the need for guidelines of biologic treatment usage in this age population and the need for additional treatment options.
Comments:
1) Abstract-line #23 dupilumab's FDA approval for adolescents (12-17y) with AD is worth mentioning in addition to EMA approvals
2)Introduction- Address insurance coverage issue with off-label treatment options which effects the decision making of physicians
3)Discussion line #230 Add citations of data on safety of biologics in psoriasis in pediatric patients:
*J Drugs Dermatol. 2015 Aug;14(8):881-6.Garber C, Systemic Treatment of Recalcitrant Pediatric Psoriasis: A Case Series and Literature Review.
*J Dermatolog Treat. 2019 Mar;30(2):152-155.Ollech A, Biologic treatment of recalcitrant pediatric psoriasis: a case series from a tertiary medical center.
4)Discussion line #231 dupilumab – emphasis that there is less data on safety in children treated with dupilumab, which is yet to be elucidated in the future.
Author Response
Abstract-line #23 dupilumab's FDA approval for adolescents (12-17y) with AD is worth mentioning in addition to EMA approvals
Response: I have included the reviewer’s suggestion in the abstract.
2)Introduction- Address insurance coverage issue with off-label treatment options which effects the decision making of physicians
Response: I have included this important topic into my introduction.
3)Discussion line #230 Add citations of data on safety of biologics in psoriasis in pediatric patients:
*J Drugs Dermatol. 2015 Aug;14(8):881-6.Garber C, Systemic Treatment of Recalcitrant Pediatric Psoriasis: A Case Series and Literature Review.
*J Dermatolog Treat. 2019 Mar;30(2):152-155.Ollech A, Biologic treatment of recalcitrant pediatric psoriasis: a case series from a tertiary medical center.
Response: I have included the reviewer’s suggested citations into the discussion.
4)Discussion line #231 dupilumab – emphasis that there is less data on safety in children treated with dupilumab, which is yet to be elucidated in the future.
Response: I have addressed this issue within the discussion in line 299 and line 233.
Reviewer 2 Report
This is a well-written article. I have to suggestions for the first paragraph in the discussion section (sentences 221-226):
Please add - Several factors must be considered while choosing the appropriate biologic therapy for psoriasis, and key considerations for the pediatric population include dosing schedule, approval for pediatric population and safety profile. Reference - Kaushik SB, Lebwohl MG. Psoriasis: Which therapy for which patient: Psoriasis comorbidities and preferred systemic agents. J Am Acad Dermatol. 2019 Jan;80(1):27-40. PubMed PMID: 30017705. Epub 2018/07/19. eng. Please mention that your analysis is in agreement with recently published recommendations for psoriasis, which concluded that etanercept and ustekinumab are preferred agents for the treatment of pediatric psoriasis and that the traditional agents should not be used for long term in the pediatric population due to the risk of side effects. Reference - Kaushik SB, Lebwohl MG. Psoriasis: Which therapy for which patient: Focus on special populations and chronic infections. J Am Acad Dermatol. 2019 Jan;80(1):43-53. PubMed PMID: 30017706. Epub 2018/07/19. eng.
Author Response
This is a well-written article. I have to suggestions for the first paragraph in the discussion section (sentences 221-226):
Please add - Several factors must be considered while choosing the appropriate biologic therapy for psoriasis, and key considerations for the pediatric population include dosing schedule, approval for pediatric population and safety profile. Reference - Kaushik SB, Lebwohl MG. Psoriasis: Which therapy for which patient: Psoriasis comorbidities and preferred systemic agents. J Am Acad Dermatol. 2019 Jan;80(1):27-40. PubMed PMID: 30017705. Epub 2018/07/19. eng.
Response: We thank the reviewer for their kind words regarding our article. We have included their important point and recommended citation into our discussion.
Please mention that your analysis is in agreement with recently published recommendations for psoriasis, which concluded that etanercept and ustekinumab are preferred agents for the treatment of pediatric psoriasis and that the traditional agents should not be used for long term in the pediatric population due to the risk of side effects. Reference - Kaushik SB, Lebwohl MG. Psoriasis: Which therapy for which patient: Focus on special populations and chronic infections. J Am Acad Dermatol. 2019 Jan;80(1):43-53. PubMed PMID: 30017706. Epub 2018/07/19. eng.
Response: We agree with the reviewer’s suggestion and have also added these details into our discussion with the recommended citation.
Reviewer 3 Report
Biological treatment plays an ever increasing role in therapies of numerous immune-mediated diseases. However, before the use of biologics, a number of conditions have to be met. The basic criterion is the lack of effectiveness of traditional treatment regimes. The rule of thumb is the assessment of advantages and disadvantages, as well as adverse effects, every time biological treatment is considered in chronic diseases.
In atopic dermatitis and psoriasis, biologics are a treatment of choice in adult patients. As the Authors rightly noticed, there is a lack of guidelines for the use of biologics in children with psoriasis and atopic dermatitis.
The Introduction is laconic, it lacks a clear clinical reference (including criteria) as to why biologics should be considered a treatment of choice in children with psoriasis and atopic dermatitis. Moreover, the Introduction does not contain a general description presenting MOA of biologics (especially in the context of psoriasis and atopic dermatitis). The lack of general references makes the paper solely a comparison of the selected medications. A concise presentation of the papers analysed should be provided in the form of tables, especially with respect to the study groups, doses, treatment duration, treatment outcomes and adverse effects.
Also, comparison of treatment outcomes and adverse effects in children and adults should be considered, which would increase the value of the subject presented.
The Discussion, just as the Introduction, is laconic. There is no reference to concomitant diseases and potential effect on ontogenesis nor identification of the role of biologics in the etiopathogenesis of diseases concomitant with psoriasis later in life (in adults).
General remarks:
- change of the type of paper into review
- change of key words (adjustment of key words to the clinical problem presented)
- since the paper is a review, the header “Results” should be removed
Conclusion: major revision
Author Response
Biological treatment plays an ever increasing role in therapies of numerous immune-mediated diseases. However, before the use of biologics, a number of conditions have to be met. The basic criterion is the lack of effectiveness of traditional treatment regimes. The rule of thumb is the assessment of advantages and disadvantages, as well as adverse effects, every time biological treatment is considered in chronic diseases.
In atopic dermatitis and psoriasis, biologics are a treatment of choice in adult patients. As the Authors rightly noticed, there is a lack of guidelines for the use of biologics in children with psoriasis and atopic dermatitis.
The Introduction is laconic, it lacks a clear clinical reference (including criteria) as to why biologics should be considered a treatment of choice in children with psoriasis and atopic dermatitis. Moreover, the Introduction does not contain a general description presenting MOA of biologics (especially in the context of psoriasis and atopic dermatitis). The lack of general references makes the paper solely a comparison of the selected medications. A concise presentation of the papers analysed should be provided in the form of tables, especially with respect to the study groups, doses, treatment duration, treatment outcomes and adverse effects.
Also, comparison of treatment outcomes and adverse effects in children and adults should be considered, which would increase the value of the subject presented.
The Discussion, just as the Introduction, is laconic. There is no reference to concomitant diseases and potential effect on ontogenesis nor identification of the role of biologics in the etiopathogenesis of diseases concomitant with psoriasis later in life (in adults).
General remarks:
- change of the type of paper into review
- change of key words (adjustment of key words to the clinical problem presented)
- since the paper is a review, the header “Results” should be removed
Conclusion: major revision
Response: We agree with the reviewer that this paper would better serve as a review and have therefore removed the header “Result”. We believe this review serves to better inform primary care providers, including pediatricians, on the use of biologics to treat pediatric and adolescent patients with psoriasis and atopic dermatitis.
We also agreed with the reviewer’s suggestion of having a table to provide a concise presentation of each biologic. We have included the mechanism of action, dosage, indications, and adverse effects for each FDA and EMA approved biologic in pediatric and adolescent populations.
We believe a comparison of treatment outcomes and adverse effects in children and adults is beyond the scope of this review, but will consider this as a future paper topic. We thank the reviewer for their suggestion.
Round 2
Reviewer 3 Report
Following the corrections the Authors have introduced I approve the article for publication.
I only sustain my previous remark about changing the paper into a Review.
This type has been clearly defined in the instructions of the journal (https://www.mdpi.com/journal/children/instructions).
I also think it would be justified to mention a slightly different causative etiology in children and first administer treatment of carries, helminthoses,sinusitis, and consider tonsillectomy before turning to biological treatment. (Childhood psoriasis.Dorota Piekarska-MyÅ›liÅ„ska1, Aldona Pietrzak1, Wojciech MyÅ›liÅ„ski1, Daniel Pietrzak1, Magdalena Borysowicz2,Mateusz Socha3, Dorota Krasowska1 Dermatol Rev/Przegl Dermatol 2017, 104, 363–376
DOI: https://doi.org/10.5114/dr.2017.69944

Author Response
Reviewer Comments: Following the corrections the Authors have introduced I approve the article for publication.
I only sustain my previous remark about changing the paper into a Review.
This type has been clearly defined in the instructions of the journal (https://www.mdpi.com/journal/children/instructions)
I also think it would be justified to mention a slightly different causative etiology in children and first administer treatment of carries, helminthoses,sinusitis, and consider tonsillectomy before turning to biological treatment. (Childhood psoriasis.Dorota Piekarska-MyÅ›liÅ„ska1, Aldona Pietrzak1, Wojciech MyÅ›liÅ„ski1, Daniel Pietrzak1, Magdalena Borysowicz2,Mateusz Socha3, Dorota Krasowska1 Dermatol Rev/Przegl Dermatol 2017, 104, 363–376
Author Reply: The reviewer makes some very excellent points. While a systematic review would provide more detail on this subject, this article serves as a literature review so that primary care providers to pediatric patients can be more familiar with biological agents. We have edited our introduction to better reflect that this paper is a literature review. Additionally, the point the reviewer brings up is an important one; however, we feel it is beyond the scope of this review.